# Increased Gene Targeting in Hyper-Recombinogenic LymphoBlastoid Cell Lines Leaves Unchanged DSB Processing by Homologous Recombination

**DOI:** 10.3390/ijms23169180

**Published:** 2022-08-16

**Authors:** Emil Mladenov, Katja Paul-Konietzko, Veronika Mladenova, Martin Stuschke, George Iliakis

**Affiliations:** 1Division of Experimental Radiation Biology, Department of Radiation Therapy, University Hospital Essen, University of Duisburg-Essen, 45122 Essen, Germany; 2Institute of Medical Radiation Biology, Medical School, University of Duisburg-Essen, 45122 Essen, Germany; 3German Cancer Consortium (DKTK), Partner Site University Hospital Essen, 45122 Essen, Germany; 4German Cancer Research Center (DKFZ), 69120 Heidelberg, Germany

**Keywords:** DT40 cells, NALM6 cells, double-strand breaks, homologous recombination, γH2AX foci, RAD51 foci, DSB repair pathway choice

## Abstract

In the cells of higher eukaryotes, sophisticated mechanisms have evolved to repair DNA double-strand breaks (DSBs). Classical nonhomologous end joining (c-NHEJ), homologous recombination (HR), alternative end joining (alt-EJ) and single-strand annealing (SSA) exploit distinct principles to repair DSBs throughout the cell cycle, resulting in repair outcomes of different fidelity. In addition to their functions in DSB repair, the same repair pathways determine how cells integrate foreign DNA or rearrange their genetic information. As a consequence, random integration of DNA fragments is dominant in somatic cells of higher eukaryotes and suppresses integration events at homologous genomic locations, leading to very low gene-targeting efficiencies. However, this response is not universal, and embryonic stem cells display increased targeting efficiency. Additionally, lymphoblastic chicken and human cell lines DT40 and NALM6 show up to a 1000-fold increased gene-targeting efficiency that is successfully harnessed to generate knockouts for a large number of genes. We inquired whether the increased gene-targeting efficiency of DT40 and NALM6 cells is linked to increased rates of HR-mediated DSB repair after exposure to ionizing radiation (IR). We analyzed IR-induced γ-H2AX foci as a marker for the total number of DSBs induced in a cell and RAD51 foci as a marker for the fraction of those DSBs undergoing repair by HR. We also evaluated RPA accretion on chromatin as evidence for ongoing DNA end resection, an important initial step for all pathways of DSB repair except c-NHEJ. We finally employed the DR-GFP reporter assay to evaluate DSB repair by HR in DT40 cells. Collectively, the results obtained, unexpectedly show that DT40 and NALM6 cells utilized HR for DSB repair at levels very similar to those of other somatic cells. These observations uncouple gene-targeting efficiency from HR contribution to DSB repair and suggest the function of additional mechanisms increasing gene-targeting efficiency. Indeed, our results show that analysis of the contribution of HR to DSB repair may not be used as a proxy for gene-targeting efficiency.

## 1. Introduction

Cells of higher eukaryotes are frequently exposed to physical and chemical agents that damage their DNA. From all types of DNA lesions induced during such exposures, the DNA double-strand breaks (DSBs) are thought to elicit the most severe cellular responses and to activate a complex signaling network termed the DNA damage response (DDR) [1,2,3]. DDR coordinates DSB repair pathways with the progression of cells through the cell cycle and safeguards genome integrity [4,5,6,7]. In cells of higher eukaryotes distinct DSB repair mechanisms have evolved with large fluctuations in accuracy, speed and effectiveness. Among them, classical non-homologous end-joining (c-NHEJ) is dominated by the enzymatic activity of the DNA-dependent protein kinase catalytic subunit (DNA-PKcs) and is characterized by a simple but fast, homology-independent rejoining of the generated ends, with half-times of approximately 10–20 minutes [8]. This form of DSB repair operates throughout the cell cycle and is considered error-prone, as there are no built-in activities ensuring sequence restoration, or the joining of the original DNA ends [9,10,11]. However, faithful restoration of DNA sequence is possible under certain circumstances, e.g. repair of restriction endonuclease induced DSBs, but is rather unlikely for IR- induced DSBs that comprise damaged nucleotides that require accurate replacement [8,12,13,14].

Homologous recombination (HR) is the only DSB repair process, which accurately restores both DNA integrity and DNA sequence at the break sites [15,16,17,18,19]. This repair mechanism is initiated by extensive DNA end-resection, resulting in the formation of 3’-single-stranded DNA (ss-DNA) overhangs, utilized in homology search during repair of the broken DNA molecules. It has been shown that the homologous sister chromatid generated after DNA replication, is the preferred template for HR, while use of the homologous chromosome is actively suppressed [20,21]. This requirement restricts HR to the S- and G_2_-phases of the cell cycle [16,22,23]. However, there is evidence that appropriate conditions may enable the initial stages of HR in G_1_-phase [20]. 

When the activities associated with c-NHEJ and/or HR are chemically or genetically compromised an alternative form of DNA end-joining (alt-EJ), also termed backup non-homologous end joining (B-NHEJ), gains ground and removes DSBs from the genome. [8,10,24,25,26]. It has been demonstrated that the activity of alt-EJ is elevated during the G_2_-phase of the cell cycle and that it is severely attenuated when cells enter a stationary phase of growth (G_0_-phase) [27,28,29,30]. The decreased fidelity of alt-EJ manifests in increased mutation induction and the accumulation of structural chromosomal abnormalities (SCAs) [31,32,33,34,35,36]. 

Another form of homology directed repair is the single-strand annealing (SSA), which is initiated when long stretches of ssDNA, generated by DNA end-resection, reveal homologies within the same DNA molecule. SSA is mostly active during the S- and G_2_-phase of the cell cycle, where DNA end-resection is at its maximum [37]. RAD52, an essential molecule in this process, catalyzes annealing between these homologous repeats, which causes the deletion of the intervening DNA segment during the subsequent repair steps [38,39,40]. Therefore, SSA is highly mutagenic [41]. 

Although DSBs challenge genomic integrity, the formation of programmed DSBs directs many physiological processes, including V(D)J and class-switch recombination, meiotic homologous recombination, activation of gene transcription and even the development of neuronal cells [42,43,44,45,46]. Therefore, it is not surprising that evolution can be driven by DSBs, and that DSB repair pathways are used by the cells to modify their genetic material by rearranging, integrating or eliminating genetic elements [47,48].

While the frequency of random genomic integration of extracellular DNA fragments is relatively high in cells of higher eukaryotes [49,50,51] and shows strong dependence on Polθ [52,53], it has been demonstrated that most somatic cell types have severe limitations in attaining effective HR-mediated integration of DNA fragments at specific genomic loci (gene-targeting). According to the principles of gene-targeting, a prerequisite for efficient targeted integration of a homologous DNA fragment is the formation of a DSB that becomes substrate of HR activities including RAD51, RAD51 paralogs, RAD54, etc. [54]. However, in higher eukaryotes, DSB repair is dominated by c-NHEJ and partially by alt-EJ, which have been suggested to suppresses HR and to limit thus gene-targeting efficiency [52,53,55]. This leads, for instance, to the high number of non-targeted integration events (1 × 10^−3^) versus targeted events (1 × 10^−6^) in HCT116 cells [51]. This is also supported by reports that only 0.09 to 0.16% of integrations in HCT116 cells, are targeted [56,57]. Although our recent results suggest a strong suppression of HR with increasing DSB load, in gene-targeting experiments typically single DSBs, flanking the homologous targeted genome sequence are introduced, ruling thus out this mechanism of HR suppression [58]. 

At the other side of the spectrum, mouse embryonic stem cells (MES) and some lymphoblastoid cell lines derived from chicken or human show increased gene-targeting efficiency suggesting increased utilization of HR that cannot be explained by a suppression of c-NHEJ or alt-EJ [50,52,53,59,60,61,62,63,64,65], and the earlier suggested contribution of topoisomerases remains unconfirmed [66,67]. Indeed, the increased gene-targeting efficiency of DT40 cells is thought to derive from the preferential use of HR in the generation of antibody diversity through gene conversion [68]. However, human NALM6 cells rely on V(D)J recombination to generate antibody diversity, which implies alternative mechanisms for their increased gene-targeting efficiency [44]. 

The increased gene-targeting efficiency of DT40 and NALM6 cells inspired us to inquire whether they also preferentially utilize HR for the repair of IR induced DSBs. Such a shift would increase HR versus c-NHEJ utilization and would suggest that similar principles underpin the phenomena of gene-targeting and DSB repair pathway choice. In the present study, we employ indirect immunofluorescence (IF) analysis of IR-induced RAD51 and γ-H2AX foci in DT40 and NALM6 cells to analyze their DSB repair characteristics and compare the results obtained to those of standard somatic cell lines – of normal tissue or tumor origin. We also adopted a DR-GFP reporter assay, as a system to measure HR activity in lymphoblastic and somatic cells. We show that the repair of IR-induced DSBs in DT40 and NALM6 cell lines fails to mirror the documented high levels of HR-mediated gene-targeting. This suggests that improved gene-targeting does not imply increased use of HR for DSB repair and indicates the operation of principles that await elucidation. 

## 2. Results

### 2.1. Undetectable Contribution of HR to DSB Repair in DT40 Cells at High IR Doses

Pulsed-field gel electrophoresis (PFGE) is a well-established and direct method for DSB repair analysis in eukaryotic cells. It measures induction of DSBs by the fraction of genomic DNA released (FDR) from cells embedded in agarose plugs into the lane of a gel subjected to an alternating electric field (Appendix A, upper panel). Therefore, FDR reflects the reduction in DNA fragment size. DSB rejoining manifests in increased size of DNA molecules and causes a decline in FDR, which is taken as evidence for DSB repair (Appendix A, down panel). It is well-documented that the numbers of IR-induced DSBs increase linearly with applied radiation dose (Appendix A) [58]. Moreover, cells of higher eukaryotes process DSBs with high efficiency throughout the cell cycle (Appendix A) [69], suggesting that the majority of DSBs are processed by a cell cycle independent repair mechanism. Since this process is strongly dependent on c-NHEJ factors, but independent of HR activities, we conclude that it mainly reflects c-NHEJ [70]. Notably, this conclusion remains similarly valid for DT40 cells (Appendix A), despite their 1000-fold increase in gene-targeting efficiency and holds also true for HR knockout DT40 cells [70]. 

To further evaluate the contribution of the cell cycle to the repair of DSBs in DT40 cells, we performed similar tests with cell cultures that have been enriched in G_2_-phase by centrifugal elutriation (Figure 1A). G_2_-phase is the phase of the cell cycle where HR is fully active. The results shown in Figure 1B,C demonstrate that G_2_-phase-enriched cells repair DSBs with efficiency similar to that of exponentially growing cells, suggesting an HR-independent DSB repair mechanism. To further confirm that this repair activity is indeed HR-independent, we introduced a small molecule inhibitor of ATR, VE-821 (to be referred as ATRi) that is known to inhibit HR [71,72]. This inhibitor suppresses RAD51 foci formation in DT40 and A549 cells (Appendix A) without affecting ATM-autophosphorylation at Serine-1981 (Appendix A).

Moreover, ATRi suppresses HR efficiency measured by the DR-GFP reporter assay in U-2 OS cells (Appendix A). Notably, even in G_2_-phase-enriched DT40 cells, ATRi as well as ATMi fail to modulate the efficiency of DSB repair (Figure 1C), which additionally confirms the limited contribution of HR to the repair of DSBs under these conditions. We conclude that at the high IR-doses required for PFGE, DSB processing is independent of gene-targeting efficiency, or HR proficiency of DT40 cells. These results are in line with the previously reported suppression of HR at high IR doses [58].

### 2.2. Contribution of HR to DSB Repair in DT40, NALM6 and Somatic Cells at Low IR Doses

We explored next, the role of HR to DSB repair in DT40 and NALM6 cells at low IR doses, where maximum contributions from HR are expected, and compared the results with similar results generated with other somatic cells. We employed immunofluorescence as previously described, to specifically analyze G_2_-phase cells [58]. NALM6 and human somatic G_2_-phase cells are identified by tracing the Cyclin B1 signal (Appendix A) [73,74]. Since this approach is not applicable in DT40 cells owing to the lack of chicken-specific antibodies, we employed centrifugal elutriation to enrich DT40 cells in G_2_-phase, as described above (Figure 1A).

We also employed quantitative image-based cytometry (QIBC) to select cells in G_1_- or G_2_-phase of the cell cycle, based on DNA content and EdU incorporation (Appendix A) [58]. EdU pulse-labelling of S-phase cells in combination with DAPI staining allows the accurate identification during analysis of cells in specific phases of the cell cycle at the time of IR. Appendix A illustrates the gates used for selection of G_1_- and G_2_-phase cells in the present set of experiments.

To monitor the induction of DSBs, we scored γ-H2AX foci formation at different doses of IR. We analyzed DT40 and NALM6 cells, as well as 82-6 hTert normal human fibroblasts and adenocarcinoma A549 cells (Figure 1D,E). The kinetics of γ-H2AX foci formation and decay shown in Figure 2A,C indicate that a dose-dependent maximum (t_max_) is reached at 1h after IR, followed at later times by a dose dependent decrease, both in G_1_- as well as in G_2_-phase cells (Figure 2A,C, and Appendix A).

Whereas 20–50 γ-H2AX foci are induced per Gy at t_max_ in human cells (fluctuations most likely reflect DNA content and cell morphology with spread out cells showing high foci numbers, and round cells, growing in suspension, showing low numbers of foci), DT40 cells, owing to their lower DNA content and round morphology, develop approximately 15 foci per Gy, 1h after irradiation. Linear induction of γ-H2AX foci is observed until foci overlapping compromises scoring with increasing IR dose (Figure 2B,D). Therefore, γ-H2AX foci at doses above 1 Gy are not taken from actual measurements, but are extrapolated from the lines shown in the corresponding Figures.

To evaluate the contribution of HR to the repair of DSBs, we also quantitated formation and decay of RAD51 foci, specifically in G_2_-phase of the cell cycle. Exposure of cells to IR, results in RAD51 foci formation in all tested cell lines (Figure 3A), suggesting that HR contributes to DSB repair in G_2_-phase; as expected RAD51 foci were undetectable in G_1_-phase cells (data not shown).

The kinetics of RAD51 foci formation follows the previously described course with a dose-dependent maximum reached at progressively longer times as the IR dose increases (Appendix A) [58]. Therefore, full kinetics were obtained to extract the number of foci at maximum, which is plotted as a function of IR dose in Figure 3B, for all cell lines tested. In contrast to the initially linear γ-H2AX induction curves, and in line with our previous report, the induction of RAD51 foci is much lower than that for γ-H2AX, which alleviates the detection problems indicated for γ-H2AX, and reaches a plateau above 2 Gy. Strikingly, DT40 and NALM6 cells display lower induction of RAD51 foci than A549 or 82-6 hTert cells, despite their dramatically higher gene-targeting efficiency. Indeed, when the fraction of induced DSBs that are processed by HR is calculated as the ratio between RAD51 and γ-H2AX foci, similar contributions are detected (Figure 4). These observations uncouple gene-targeting efficiency from HR contribution to the repair of randomly generated DSBs.

### 2.3. Analysis of DNA End Resection in NALM6 and A549 Cells

A major decision in DSB repair pathway choice and HR engagement is the inception of resection at DSB ends. Therefore, we quantitated resection to examine possible differences between NALM6 and A549 cells. The quantification of RPA accumulation at ssDNA is an established marker of resection that can be quantitated using bi-variate flow cytometry, specifically in G_2_-phase cells exposed to 5 or 10 Gy of X-rays (Figure 5A,B, and Appendix A for details of analysis). The robust increase in RPA signal measured after irradiation documents resection and demonstrates that also at this end point, NALM6 cells fail to show enhanced responses as compared to A549 cells (Figure 5B).

We further inquired whether the higher targeting efficiency in NALM6 cells derives from altered balance between c-NHEJ and HR. Therefore, we analyzed levels of key proteins of c-NHEJ and HR. Western blot analysis shows (Figure 5C) that the levels of RAD51, KU70, KU80 and RPA32 are similar between tested cell lines, thus eliminating the difference in protein levels as pivotal for the response [75,76].

### 2.4. RAD51 Foci Formation in DT40 Mutants with Various Repair Defects

We also examined in DT40 cells how different defects in components of proteins linked to different DSB repair pathways affect the dose response of RAD51 foci induction (Figure 6A,B). The rational and procedures used in these experiments are identical to those outlined for the results in Figure 3. It is evident that while DNAPKcs defects increase RAD51 foci induction with a pronounced difference at high-doses, ATM defects generate only marginal effects, while defects in components of the HR apparatus such as, LIG1, MRE11 and CtIP, markedly reduce RAD51 foci formation (Figure 6A,B). One caveat in all analyses in DT40 and NALM6 cells is that their strongly pro-apoptotic phenotype requires the use of apoptosis inhibitors. Although not evident in the experiments described here, these inhibitors may generate side-effects confounding the results obtained.

### 2.5. Comparison of HR Efficiency in DT40 and U2OS Cells Using the DR-GFP Reporter Assay

We adopted an established GFP reporter assay, capable of evaluating in vivo the efficiency of HR events at I-SceI-generated DSBs, within the DR-GFP gene cassette that is stably integrated in their genomes [77] (Appendix A). We used an established reporter cell line generated in U-2 OS cells as a representative of somatic cells. We also generated a transgenic DT40 (DR-GFP) cell line, by random integration of the DR-GFP cassette into the DT40 genome (similar experiments failed to produce useful clones in NALM6 cells). The successful integration of the DR-GFP vector in the puromycin selected clone, is indicated by the fact that its doubling time in the presence of puromycin is similar to the doubling time of parental DT40 cells (data not shown), and by the fact that appropriately selected PCR primers amplify the expected region of the integrated GFP gene (Appendix A).

The transfection efficiency of DT40 cells is monitored by electroporation of cells with EGFP-N1 plasmid (Appendix A) using two transfection programs optimized either for transfection efficiency, or for cell viability (Appendix A). Twenty-four hours after induction of DSBs by transfection with the I-SceI expression plasmid, the number of GFP positive DT40 cells increases from 0.02% in the mock-transfected samples to about 7% in I-SceI transfected cells, which is an indication of successful HR (Figure 7A and Appendix A). Similar results are obtained with U-2 OS (DR-GFP) cells, where the specificity of the reaction for HR is also confirmed by RAD51 knock-down (Appendix A).

DT40 (DR-GFP) and U-2 OS (DR-GFP) cells show after post-transfection treatment with ATRi or ATMi a marked reduction in HR, suggesting the operation of similar mechanisms in the two cell lines (Figure 7 and Appendix A). Importantly, treatment of I-SceI transfected cells with DNA-PKcsi, increases the incidence of HR in both cell lines, confirming the reported suppression of HR by c-NHEJ (Figure 7 and Appendix A).

## 3. Discussion

Altered DSB processing in higher eukaryotes confers a dramatic increase in the random integration of DNA fragments and causes a severe reduction in gene-targeting efficiency, as compared to lower eukaryotes [78]. There are however exceptions to this rule and the lymphoblastoid cell lines NALM6 and DT40 are two of the better-known examples of cell lines with nearly 1000-fold increased gene-targeting efficiency [62,79]. Whereas the dominant role of HR in antibody maturation by immunoglobulin gene conversion provides a rationale for increased gene-targeting in DT40 cells [68,80], the underpinning of the gene-targeting increase in NALM6 cells remain unknown, and in both cell lines cannot be attributed to a suppression of Polθ dependent alt-EJ or c-NHEJ; as the expression of proteins relevant to these repair pathways remain at normal levels [52,53].

In the present work we inquired whether the mechanisms underpinning the increased gene targeting efficiency of DT40 and NALM6 cells have consequences to the repair of randomly induced DSBs in the genomes of cells exposed to IR - for instance by enhancing the utilization of HR. Therefore, we analyzed the fraction of DSBs processed by HR after exposure to a range of IR doses, and compared the results to those obtained with widely used normal or tumor cell lines. The total load of induced DSBs was determined by γ-H2AX foci analysis, whereas the subset of DSBs processed by HR was measured using RAD51 foci analysis. Thus, we were able to use the ratio between maximally induced RAD51 and γ-H2AX foci, as a measure of HR engagement. To our surprise, we found that DT40 and NALM6 cells utilize HR to a similar extend as other somatic cells (Figure 4). This similarity persisted even at very low doses of IR, where our earlier work [58] showed significant increases in HR engagement. Notably, these results extend our initial observations of HR suppression with increasing DSB load in DT40 and NALM6 cells, emphasizing the rather general character of the effect [58]. 

The number of γ-H2AX and RAD51 foci in DT40 and NALM6 cells, were two to three times lower than in 82-6 hTert and A549 cells, possibly reflecting, at least partly, their different morphology. In addition, DT40 cells have a documented genome size of 1.072 × 10^9^ base pairs, which is almost 3 times smaller than the human genome (~3 × 10^9^) [81]. The lower number of γ-H2AX foci detected in DT40 cells is contrasted by an increase in foci-size, suggesting differences in the inception of DDR and possibly also in chromatin organization. The role of chromatin organization in this response is further supported by the reported differences in condensin accumulation at specific genomic locations between chicken and human cells [82,83]. On the other hand, the mechanisms underlying the lower γ-H2AX foci yields in NALM6 cells remain unknown and will require further investigations on the nature of DDR and the organization of chromatin in these cells. 

DSB repair by HR requires resection of DSB-ends, a process mainly catalyzed by CtIP, the MRN complex, DNA2/BLM and EXO1. It was conceivable that this process was specifically modified in NALM6 cells to support their increased gene-targeting efficiency. To our surprise, again, resection was very similar in NALM6 and A549 cells, ruling thus out this initial HR step as the key facilitator of the increased gene-targeting. Moreover, the similarity in the engagement of HR, between NALM6 and other somatic cell line is also supported at the biochemical level, by the similar expression patterns of key HR proteins.

To exclude that altered dynamics of RAD51 and γ-H2AX foci formation and decay mask actual increases in HR repair in the tested cells, we integrated a widely used DR-GFP reporter substrate in DT40 cells, thus generating a lymphoblastic cell line allowing the analysis of HR efficiency. The DR-GFP reporter assay demonstrated that the fraction of GFP-positive DT40 cells after generation of DSBs at I-*SceI* sites, was similar to the number of GFP positive cells measured in U-2 OS cells. In order to confirm our interpretation of the above results, we incorporated the DR-GFP reporter construct into the genome of human NALM6 cells (data not show). Although the results indicated that repair of DSBs by HR is regulated similarly as in U-2 OS (DR-GFP) cells, the number of GFP-positive cells was extremely low, preventing firm conclusions. We conclude that even assays evaluating the efficiency of the global HR, fail to uncover differences in the processing of I-*SceI* induced DSBs, thus failing to explain the increase in gene-targeting efficiency.

Collectively the above results fail to identify a plausible culprit that would explain the increased targeted integration of external DNA in NALM6 and DT40 cells. While altered balance in DSB repair pathway selection remains a possible contributor, other mechanisms are likely to additionally play a pivotal role. Our efforts at present focus on the role of differences in chromatin organization [84,85,86]. Differences in chromatin landscape, particularly the balance between heterochromatin and euchromatin content, which are known to influence DDR and DNA repair [84,85,86], may also underpin the observed variations between lymphoblastic and other somatic cell lines. It is possible that in the cell lines tested, chromatin is more accessible to targeting vectors. Such a model is supported by data from CRISPR/Cas12a DNA targeting, which was shown to be inhibited by nucleosomes and chromatin compaction [87]. Further evidence on a role of chromatin on HR-mediated gene targeting efficiency comes from human cells transfected with a construct of a chimeric Cas9 fused to PRDM9 chromatin remodeling factor [88]. This factor deposits histone methylation marks (H3K4me3 and H3K36me3) that facilitate HR, and causes a 3-fold increase in HR-mediated gene targeting versus indels formation [88].

In summary, our observations allow us to conclude that the increased gene-targeting efficiency of DT40 and NALM6 cells fails to be explained by increases in the individual steps of HR that also benefit repair by HR of randomly induced DSBs. While the results obtained in the current study are short of offering a mechanistic explanation on the underpinning mechanisms, they identify for the first time the dichotomy between HR utilization for DSB repair and gene targeting efficiency. We postulate that the increased gene targeting efficiency of NALM6 and DT40 cells derives from altered chromatin structure that facilitates the exposure of homologous sequences to the gene-targeting vectors. We hope that the reported negative results will be useful to the field and that they will inspire research on the underpinning molecular mechanisms. This information may ultimately guide developments improving gene targeting and thus gene therapy.

## 4. Materials and Methods

### 4.1. Cell Lines and Culture Conditions

The Null Acute Leukemia Minowada 6 (NALM6) cells were cultivated in Roswell Park Memorial Institute 1640 medium (RPMI) with 10% fetal bovine serum (FBS). DT40 cells were grown in DMEM/F12 media at 41 °C. The 82-6 hTert human fibroblasts is an hTert-immortalized cell line, derived from a normal, 82-6, fibroblast cells. 82-6 hTert cells were grown in MEM media supplemented with 10% FBS and 1% non-essential amino acids at 37 °C. A549 and U-2 OS cells are, respectively, pulmonary and osteosarcoma derived cell lines and were grown in Mc Coy’s 5A media supplemented with 10% FBS at 37 °C. All cell growth media were supplemented with Penicillin/Streptomycin antibiotics and all cells were grown in the presence of 5% CO_2_ and 95% air.

### 4.2. Irradiation Conditions

NALM6 and DT40 cells were treated 1h prior to IR with 100 µM Caspase-3 inhibitor Boc-D-FMK (Calbiochem, CAS#:634911-80-1). Cells were irradiated in an Isovolt-320 kV X-ray tube (GE-Healthcare) operating at 320 kV, 10 mA, with a 1.65 mm aluminum filter. Irradiations were carried out at room temperature at a distance of 500 mm. The effective photon energy was ~70–90 keV and the dose rate ~3.25 Gy/min. 

### 4.3. Indirect Immunofluorescence (IF) and Quantitative Image-Based Cytometry (QIBC) Analysis of γ-H2AX and RAD51 Foci

After irradiation, NALM6 and DT40 cells were attached to especially coated, adherent slides (Carl ROTH) for 10 min on ice. When adherent cells (82-6 hTert and A549) were used, cells were grown directly on cover slips and were irradiated as described above. At the indicated times after irradiation, cells were fixed in 3% paraformaldehyde solution, containing 2% sucrose for 15 min at room temperature, permeabilized in P-Solution (50 mM EDTA, pH 8.0, 50 mM Tris-HCl, pH 7.6), containing 0.5% Triton X-100 and blocked in PBG buffer (0.2% skin fish gelatin, 0.5% BSA in PBS) overnight at 4 °C. Cells were incubated with monoclonal primary antibodies against γ-H2AX (Abcam, Cambridge, UK, clone [3F2]) or RAD51 (GeneTex, Irvine, CA, USA, clone [14B4]), together with a polyclonal Cyclin B1 antibody (Santa Cruz Biotechnology, Inc., Dallas, TX, USA, H-300) for 1.5 h at room temperature and after 3 washes with PBS, were incubated for an additional 1.5 h with the corresponding secondary antibodies, conjugated with AlexaFluor 488 or AlexaFluor 568. Nuclei were counterstained with DAPI. Immunofluorescence images were taken by a Leica SP-5 confocal microscope. γ-H2AX and RAD51 staining were accomplished within the same experiment.

For QIBC analysis cells were grown on coverslips and 30 minutes before IR were pulse-labeled with 2 μM of 5-ethynyl-2′-deoxyuridin (EdU), which labels S-phase cells. Immediately thereafter, cells were irradiated and EdU was washed out by exchanging growth medium with fresh EdU-free growth medium. Cells were collected for IF analysis as described above. The EdU signal was developed using an EdU staining kit, (Click-It, Thermo Fisher Scientific, Waltham, MA, USA), according to the manufacturer’s instructions. AxioScan.Z1 (Carl Zeiss Microscopy) was utilized to scan selected areas of 4 × 4 mm, containing approximately 10,000–20,000 cells. QIBC analysis combining EdU and DAPI signals allowed us to discriminate the cell cycle phase in which cells were at the time of irradiation. Cellular segmentation analyses and γ-H2AX or RAD51 foci in the same experiment were quantified by using the Imaris 8.0-9.5.1 software (Bitplane) and the generated data converted into the proper format for utilization with a flow cytometry software for presentation and analysis purposes (Kaluza 2.1; Beckman Coulter, Bream, CA, USA; Krefeld, Germany).

### 4.4. RNA Interference

siRNA against RAD51 was from Qiagen as FlexiTube siRNA, Hs_Rad51_7. About 2–3 × 10^6^ U-2 OS cells harboring DR-GFP reporter substrates [71] were transfected by nucleofection with 5 μL of 20 μM siRNA stock solution in 100 μL transfection buffer using Nucleofector^tm^ 2d device (LONZA). Twenty-four hours after transfection, cells were collected and subject to nucleofection with I-SceI-expression plasmid (see next section). 

### 4.5. DSB Repair in the Intrachromosomal DR-GFP Reporter Substrate

About 2 × 10^6^ U-2 OS or DT40 cells, harboring the DR-GFP reporter construct [77] were transfected with 1 μg I-*SceI* expressing plasmid and were treated, or not, with 5 μM of the specific ATR inhibitor VE-821 (ATRi), or 10 μM of the ATM inhibitor, KU55933 (ATMi), or 10 μM of the DNA-PKcs inhibitor NU7441 (DNAPKcsi). Twenty-four hours later cells were analyzed for GFP expression by flow cytometry. The number of GFP positive cells (GFP+) determined in depleted or inhibitor-treated cells was expressed as a percentage of GFP+ cells measured in mock-transfected control cells.

### 4.6. Generation of A DT40 (DR-GFP) Reporter Cell Line

About 10 × 10^6^ wild-type DT40 cells were transfected by nucleofection with a XhoI linearized plasmid harboring the DR-GFP reporter construct. Twenty-four hours after nucleofection, one million cells were seeded in triplicates on methylcellulose plates containing 2 μg/mL puromycin and the dishes were incubated for 10 days at 41 °C, in a CO_2_ incubator. Single colonies were isolated and tested for positive integration of the DR-GFP cassette by PCR (Appendix A) and for HR functionality by measuring GFP-positive cells using FACS after I-*SceI* transfection. Similar efforts failed to generate useful clones in NALM6 cells.

## Figures and Tables

**Figure 1 ijms-23-09180-f001:**
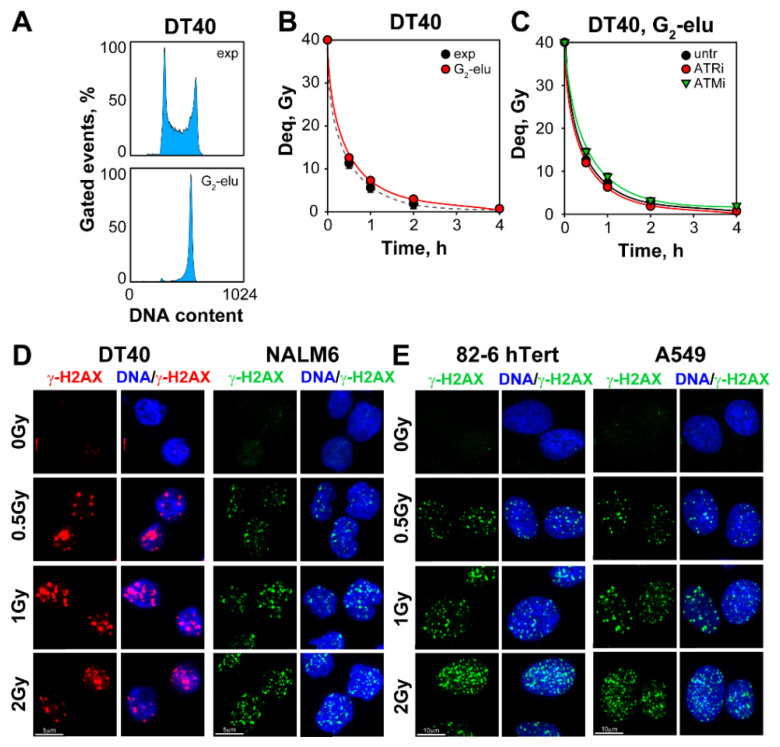
Contribution of HR to the repair of DSBs is undetectable by PFGE at high IR-doses: (**A**) Histogram plots representing the cell cycle distribution of DT40 cells before (exp) and after centrifugal elutriation (G_2_-elu); (**B**) Repair kinetics of exponentially growing cells (exp) and cells enriched in G_2_-phase of the cells cycle (G_2_-elu), irradiated with 40 Gy and analyzed by PFGE; (**C**) Repair kinetics of G_2_-phase-enriched DT40 cells, pre-treated or not with ATRi or ATMi; (**D**) IF images of γ-H2AX foci formation in DT40 and NALM6 cells, irradiated with the indicated doses of IR. Images are captured by an AxioScan Z1 platform and are utilized to generate the QIBC analysis data; (**E**) Same as Figure 1D, but for 82-6 hTert fibroblasts and A549 lung cancer cells.

**Figure 2 ijms-23-09180-f002:**
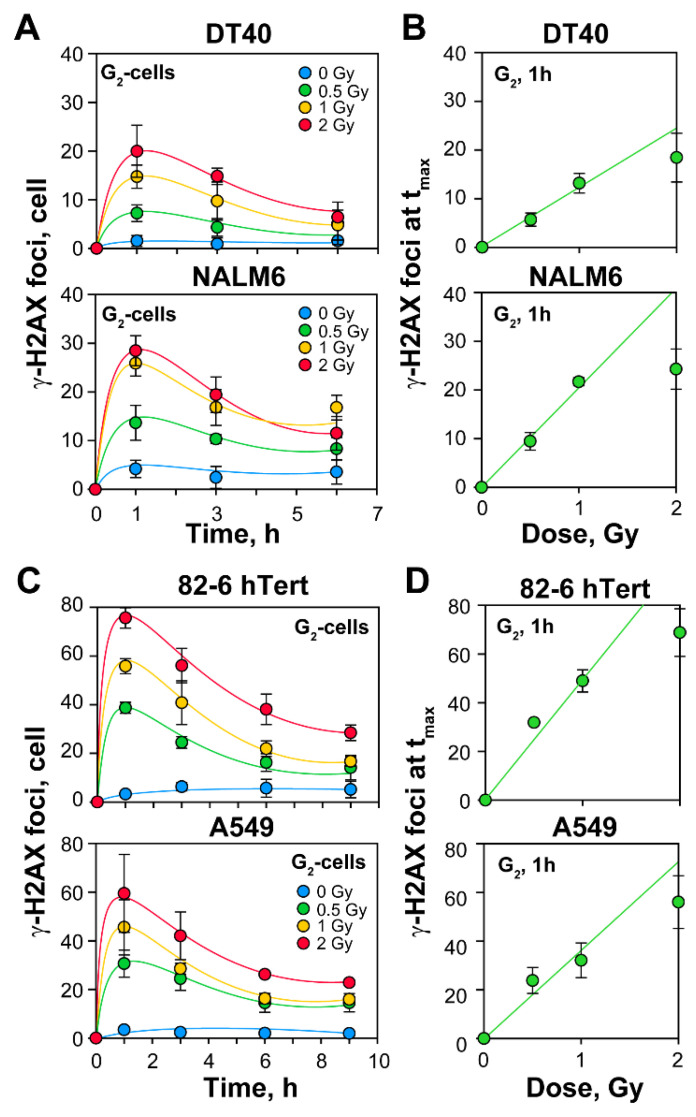
Formation of γ-H2AX repair foci indicates a linear increase of DSBs with increasing radiation dose: (**A**) Repair kinetics of γ-H2AX foci formation and decay in DT40 and NALM6 cells; (**B**) Quantitative analysis of γ-H2AX foci in lymphoblastoid cell lines; (**C**) Same as in Figure 2A, but for human 82-6 hTert and A549 cell lines; (**D**) Same as in Figure 2B, but for human 82-6 hTert and A549 cell lines. Dose response curves are derived from the repair kinetic experiments shown in Figure 2A,C and represent the formation of γ-H2AX foci in G_2_-phase cells, 1h after IR exposure (t_max_). Data is generated by QIBC analysis and represent the mean and standard deviation from 3 independent experiments.

**Figure 3 ijms-23-09180-f003:**
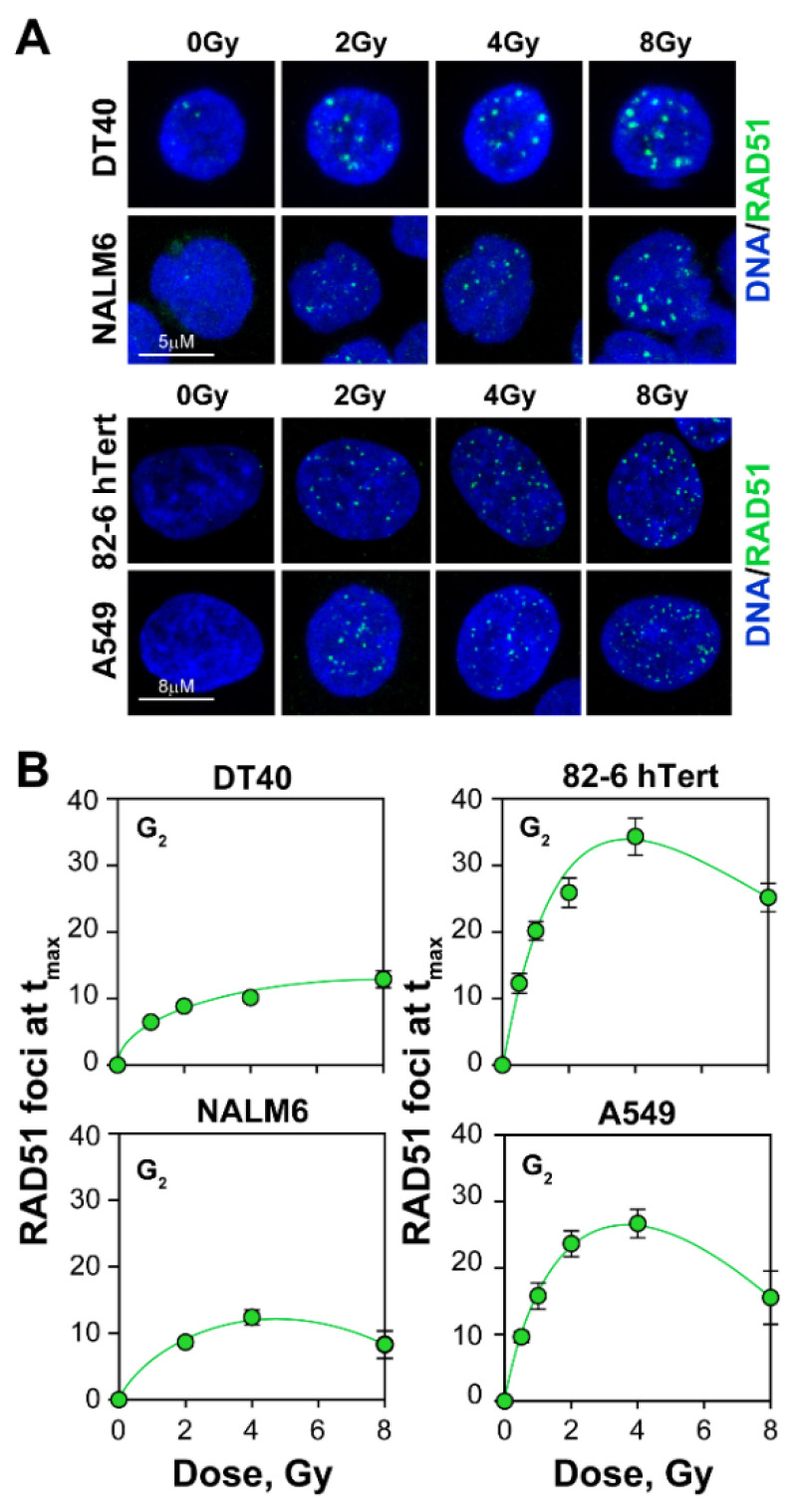
Formation of RAD51 foci, a marker for ongoing HR repair, at DSBs is suppressed at high radiation doses: (**A**) Representative IF images of RAD51 foci in DT40, NALM6, 82-6 hTert and A549 cells; (**B**) QIBC analysis of RAD51 foci in lymphoblastoid and other somatic cell lines. The dose response curves are derived from the repair kinetic experiments (Appendix A) and represent the formation of RAD51 foci at their dose-specific maximum (t_max_). Data indicate the mean and standard deviations from 3 independent experiments.

**Figure 4 ijms-23-09180-f004:**
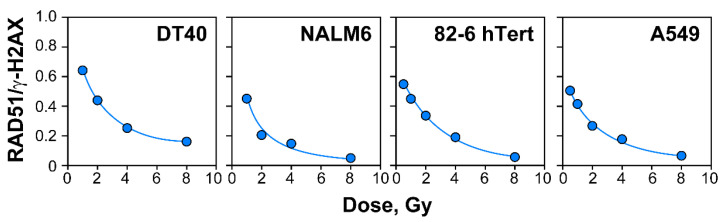
The contribution of HR to the repair of DSBs decreased with an increasing radiation dose in both lymphoblastic and other somatic cell lines. Plots illustrating the ratio between RAD51 foci, marking the number of DSBs repaired by HR and γ-H2AX foci, as an indicator of the overall DSB load. The numbers of γ-H2AX foci at radiation doses above 1 Gy were extrapolated from the fitted curves shown in Figure 2B,D, while the number of RAD51 foci at (t_max_) was determined from the experiments shown in Figure 3 and Appendix A.

**Figure 5 ijms-23-09180-f005:**
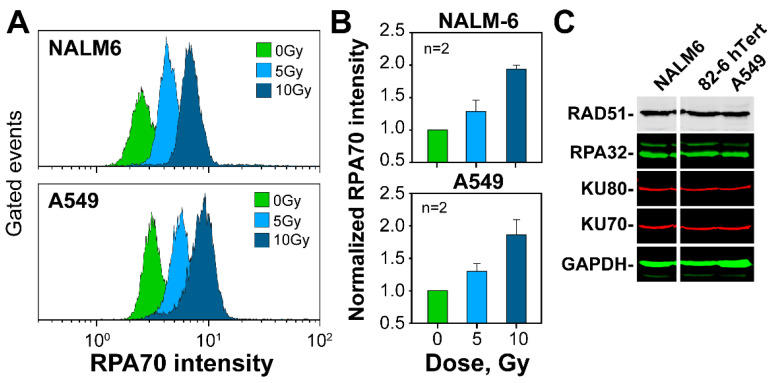
NALM6 and A549 cells show similar levels of DNA end-resection during DSB processing: (**A**) Representative flow cytometry histogram plots indicating the distribution of RPA70 intensity in non-irradiated cells (0Gy) and cells irradiated with 5 or 10 Gy of X-rays. Only G_2_-phase cells are included in the analysis. The gates used for the selection of G_2_-phase cells are shown in Appendix A; (**B**) Quantification of the histogram plots shown in Figure 5A; (**C**) Western blot analysis in NALM6, 82-6 hTert and A549 cell lines, of protein factors involved in the repair of DSBs. Uncropped Western blot membranes are shown in Appendix A.

**Figure 6 ijms-23-09180-f006:**
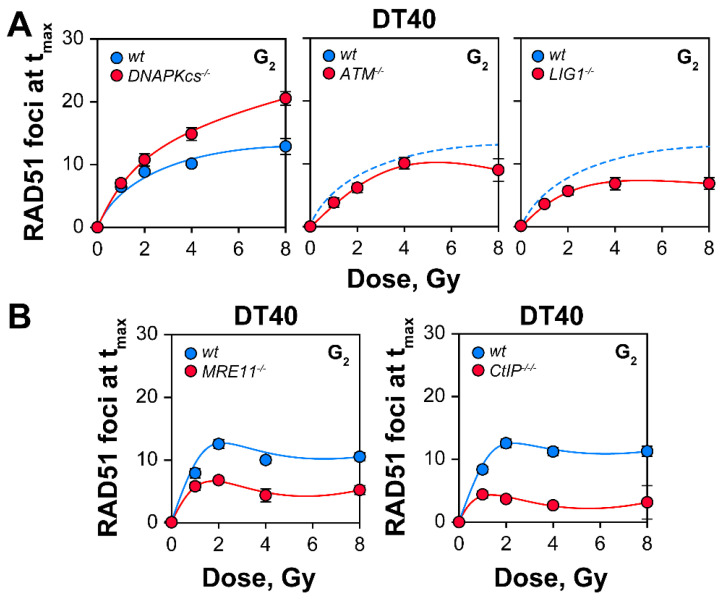
RAD51 foci in DT40 cells with various DSB repair defects: (**A**) Formation of RAD51 foci in DT40 cells deficient in factors involved in DSB-processing (DNAPKcs, ATM and LIG1); (**B**) RAD51 foci in DT40 cells deficient in the essential factors involved in DNA end-resection (MRE11 and CtIP). G_2_-phase cells were enriched by centrifugal elutriation. Data represents the mean and standard deviations from 3 independent determinations.

**Figure 7 ijms-23-09180-f007:**
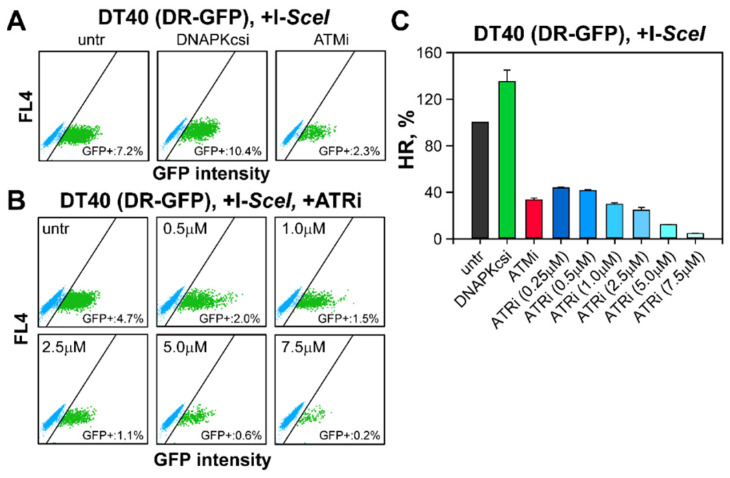
Repair of I-*SceI* induced DSBs in DT40 cells is comparable to DSB repair in U-2 OS, DR-GFP reporter cells: (**A**) Representative dot plots of GFP positive cells in DT40 (DR-GFP) cells. After transfection with I-*SceI*-expressing plasmids, cells were grown either in inhibitor-free media or in media containing DNA PKcs and ATM inhibitors. The GFP signal was analyzed 24 h after transfection; (**B**) Representative dot plots of GFP positive cells in DT40 (DR-GFP) clone, after treatment with increasing concentrations of ATRi; (**C**) Quantitative analysis of GFP positive cells in the experiments shown in Figure 7A,B. The percentage of GFP positive cells reflects DSB repair by HR (HR, %). HR efficiency in cells treated with PIKK inhibitors is calculated by normalizing to the levels of untreated cells. Data represents the mean and standard deviation from two experiments.

## Data Availability

All data will be available upon request to the corresponding author.

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
