# Peer review of "Increased Gene Targeting in Hyper-Recombinogenic LymphoBlastoid Cell Lines Leaves Unchanged DSB Processing by Homologous Recombination"

_ijms, 2022, doi:10.3390/ijms23169180_

Round 1

Reviewer 1 Report

Mladenov et al. examined whether the high gene-targeting efficiency of chicken DT40 or human NALM6 cell line is linked to a high rate of HR repair of DSBs which was a reasonable hypothesis.  They monitored IR-induced gamma-H2AX, RAD5, and RPA foci as proxies of DSBs, HR repair, and DNA end resection  in DT40 and NALM6 vs. a few other human cell lines with low gene targeting efficiency. Previously established methods were to conduct the experiments, each of which yielded fairly clear results. Their results did not seem to support the aforementioned  hypothesis. They posited that high gene targeting is a result of altered chromatin structure in DT40 and NALM6, but didn’t discuss it in any depth.

Notably, to estimate the contribution of HR to DSB repair, the authors calculated the ratio of the number of RAD51 foci over that of gamma-H2AX foci (as a measure of HR engagement). While this procedure was likely more accurate than counting RAD51 foci alone, the authors monitored RAD51 and H2AX foci in different cells in separate experiments for any given cell line, compromising the value of this experiment.

Regarding the DR-GFP recombination assay, the authors compared chicken DT40 to human U2OS, which is not desirable. It’s better to monitor DR-GFP recombination in NALM6 vs. U2OS.

Author Response

Response to Reviewer Comments

Reviewer 1

  1. “Mladenov et al. examined whether the high gene-targeting efficiency of chicken DT40 or human NALM6 cell line is linked to a high rate of HR repair of DSBs, which was a reasonable hypothesis.”  

We thank the Reviewer for appreciating the significance and the rationale of our work.

  1. “They monitored IR-induced gamma-H2AX, RAD52, and RPA foci as proxies of DSBs, HR repair, and DNA end resection in DT40 and NALM6 vs. a few other human cell lines with low gene targeting efficiency.”

The outline by the Reviewer accurately describes the approach taken in the present study.

  1. “Previously established methods were used to conduct the experiments, each of which yielded fairly clear results.”

We very much appreciate that the Reviewer recognizes the clarity of our results, which of course strengthens our conclusions.

  1. “Their results did not seem to support the aforementioned hypothesis. They posited that high gene targeting is a result of altered chromatin structure in DT40 and NALM6, but didn’t discuss it in any depth.”

The Reviewer identifies an important issue with the paper, which we appreciate very much. In the revised version we have expanded the Discussion to provide more depth regarding the potential role of chromatin organization and DSB repair pathway balance in gene targeting.

  1. “Notably, to estimate the contribution of HR to DSB repair, the authors calculated the ratio of the number of RAD51 foci over that of gamma-H2AX foci (as a measure of HR engagement). While this procedure was likely more accurate than counting RAD51 foci alone, the authors monitored RAD51 and H2AX foci in different cells in separate experiments for any given cell line, compromising the value of this experiment.”

This is also a relevant point and we appreciate the concern of the Reviewer. We clarify at the outset and mention in the revised manuscript under “Materials and Methods” that gH2AX and RAD51 foci were typically scored together in the same experiment, albeit in different pairwise combinations to accommodate antibody properties and dye separation. Moreover, since our analysis is highly cell-cycle-specific (focusing on G2-phase cells) and includes the analysis of many cells, we can be reasonably confident that our results reflect well our conclusions.

  1. “Regarding the DR-GFP recombination assay, the authors compared chicken DT40 to human U2OS, which is not desirable. It’s better to monitor DR-GFP recombination in NALM6 vs. U2OS.”

The Reviewer is certainly right in his suggestion and indeed we attempted this. However, our attempts to generate DR-GFP homologous recombination reporter NALM6 cells resulted in clones showing extremely low frequency of GFP positive cells upon transfection with an I-SceI expressing plasmid, which seriously compromised their utility. Indeed, the generation and validation of such reporter cell clones in different cell lines is highly time-consuming and of uncertain outcome. In this regard, it is actually highly indicative that despite the extensive use of these reporter assays in the field to analyze DSB repair pathway activity, the vast majority of studies, use practically exclusively, the cell lines generated by Dr. Jeremy Stark – as we partly do too.

Nevertheless, as a compromise, in the revised version of the manuscript, we have included an experiment investigating the expression patterns of key DSB repair proteins in lymphoblastic, NALM6 and somatic cells. The results revealed no detectable differences in the levels of RPA32, RAD51, KU70 and KU80 proteins in the investigated cell lines, limiting thus further the potential parameters underpinning the increased gene targeting efficiency in DT40 and NALM6 cells.

Reviewer 2 Report

This manuscript titled with “Increased gene-targeting in hyper-recombinogenic lymphoblastoid cell lines leaves unchanged DSB processing by homologous recombination”, or IJMS-1829098, was truly scholarly carried out and it went through basic properties to evaluate HR efficiency in DT40 and NALM6 cell lines, in comparison to other somatic cell lines. While I personally enjoyed reading this manuscript, I believe this manuscript did not offer the audiences solid positive conclusions, with regards to the previous observation of elevated gene-targeting efficiency. As a conclusion, this is a manuscript with only negative conclusions. Because of so, I cannot expect any efforts to be made in 2-3 months to revise this current manuscript to the degree that a positive conclusion is reached.

Author Response

Response to Reviewer Comments

Reviewer 2

  1. “This manuscript titled with “Increased gene-targeting in hyper-recombinogenic lymphoblastoid cell lines leaves unchanged DSB processing by homologous recombination”, or IJMS-1829098, was truly scholarly carried out and it went through basic properties to evaluate HR efficiency in DT40 and NALM6 cell lines, in comparison to other somatic cell lines.”

We thank the Reviewer for this highly positive assessment of our work.

  1. “While I personally enjoyed reading this manuscript, I believe this manuscript did not offer the audiences solid positive conclusions, with regards to the previous observation of elevated gene-targeting efficiency. As a conclusion, this is a manuscript with only negative conclusions. Because of so, I cannot expect any efforts to be made in 2-3 months to revise this current manuscript to the degree that a positive conclusion is reached.”

Again, we very much thank the Reviewer for this positive statement regarding our manuscript. We wish to emphasize however that the aim of our work was not to analyze the increased gene targeting efficiency of DT40 and NALM6 cell lines, but to investigate whether this increase is associated with increases in HR engagement on radiation-induced DNA double strand breaks. Our results demonstrate that such association is unlikely to exist, which is a positive outcome that should be of interest to those working in the field, as it informs experiments and models that go beyond the HR machinery. While the results obtained are short of offering a mechanistic explanation on the underpinning mechanisms, they identify the dichotomy between HR utilization for DSB repair and gene targeting efficiency for the first time. As such, it merits in our opinion, publication to alert other investigators in the field to accordingly focus their research efforts in other directions.

We agree with the Reviewer that it is unlikely that we will be able to provide answers to this rather complex question in a few months and indeed real breakthroughs may take years. We believe however that the effect is important and relevant and that it will attract other investigators in the field. As such the paper may be significant to the field based on its potential to ignite new research. We hope that the Reviewer sees merit in this assessment.

Round 2

Reviewer 1 Report

The authors' response to my comments is mostly satisfactory. 

Their reply to the point “Regarding the DR-GFP recombination assay, the authors compared chicken DT40 to human U2OS, which is not desirable. It’s better to monitor DR-GFP recombination in NALM6 vs. U2OS.”  is acceptable albeit it doesn't completely resolve the issue. The authors should mention the potential limitation of their DR-GFP experiment in the paper. 

Reviewer 2 Report

I will let it go this time but I insist that this is a negative result manuscript. 

Author Response

Response to Reviewer Comments

Reviewer 2

  1. “I will let it go this time but I insist that this is a negative result manuscript.”

We thank again, the Reviewer for his positive decision. In the re-revised version of the manuscript we state at the closing paragraphs of the manuscript the nature of the summarized results and that the underpinning mechanisms will require additional studies.